# The HPAfrica protocol: Assessment of health behaviour and population-based socioeconomic, hygiene behavioural factors - a standardised repeated cross-sectional study in multiple cohorts in sub-Saharan Africa

Gi Deok Pak,[1] Andrea Haekyung Haselbeck,[1] Hyeong Won Seo,[1] Isaac Osei,[2] John Amuasi,[2,3] Robert F Breiman,[4] Ligia Maria Cruz Espinosa,[1] Marianne Holm,[1] Justin Im,[1] Geun Hyeog Jang,[1] Hyon Jin Jeon,[1] Stephen P Luby,[5] Octavie Lunguya-Metila,[6,7] William MacWright,[4] Ondari Daniel Mogeni,[1] Iruka N Okeke,[8] Ellis Owusu-Dabo,[3] Jin Kyung Park,[1] Se Eun Park,[1] Oluwafemi Popoola,[8] Hye Jin Seo,[1] Abdramane Bassiahi Soura,[9] Mekonnen Teferi,[10] Trevor Toy,[1] Yun Chon,[1] Mathilde Rafindrakalia,[11] Raphaël Rakotozandrindrainy,[11] Christian G Meyer,[12,13] Florian Marks,[1,14] Ursula Panzner[1,15,16]

GDP and AHH contributed equally.

For numbered affiliations see end of article.

**Correspondence to**
Dr Florian Marks;
fmarks@ivi.INT

## ABSTRACT

**Introduction** The objective of the Health Population Africa (HPAfrica) study is to determine health behaviour and population-based factors, including socioeconomic, ethnographic, hygiene and sanitation factors, at sites of the Severe Typhoid Fever in Africa (SETA) programme. SETA aims to investigate healthcare facility-based fever surveillance in Burkina Faso, the Democratic Republic of the Congo, Ethiopia, Ghana, Madagascar and Nigeria. Meaningful disease burden estimates require adjustment for health behaviour patterns, which are assumed to vary among a study population.

**Methods and analysis** For the minimum sample size of household interviews required, the assumptions of an infinite population, a design effect and age-stratification and sex-stratification are considered. In the absence of a population sampling frame or household list, a spatial approach will be used to generate geographic random points with an Aeronautical Reconnaissance Coverage Geographic Information System tool. Printouts of Google Earth Pro satellite imagery visualise these points. Data of interest will be assessed in different seasons by applying population-weighted stratified sampling. An Android-based application and a web service will be developed for electronic data capturing and synchronisation with the database server in real time. Sampling weights will be computed to adjust for possible differences in selection probabilities. Descriptive data analyses will be performed in order to assess baseline information of each study population and age-stratified and sex-stratified health behaviour.

### Strengths and limitations of this study

► Standardisation in community-based multicountry/multisite research requires cautiousness considering variations in setting, language and culture.
► Large-scale public health assessments will complement disease burden data by investigating sources of transmission and infection.
► Current population data in study sites and their administratively defined subareas and population sampling frames are assumed to be limited.
► Electronic data collection may cause technical problems related to damage, functional failure or loss of devices.
► Unstable wireless internet connection may limit data teams to update the Android-based application, communicate with study managers in real time and synchronise collected data with the server.

This will allow adjusting disease burden estimates. In addition, multivariate analyses will be applied to look into associations between health behaviour, population-based factors and the disease burden as determined in the SETA study.

**Ethics and dissemination** Ethic approvals for this protocol were obtained by the Institutional Review Board of the International Vaccine Institute (No. 2016–0003) and by all collaborating institutions of participating countries. It is anticipated to disseminate findings from this study through publication on a peer-reviewed journal.

## INTRODUCTION

Assessment of health behaviour, including healthcare utilisation, is an important determinant for the generation of accurate disease burden estimates among target populations. This is particularly critical for studies which use observational, healthcare facility-based surveillance designs as substantial variation in health behaviour patterns strongly influences disease burden estimates.[1] The adjustment of disease burden calculations for variation in health behaviour will increase the accuracy of measures of disease occurrence caused by bacterial, fungal, viral and parasitic pathogens among populations under investigation. Furthermore, the influence of a large variety of population-based factors on both health behaviour and disease burden has been recognised. This includes socioeconomic factors such as education, occupation, income, the availability of household assets,[2 3] ethical/religious and cultural factors,[4 5] and hygiene and sanitation facilities,[6–10] food handling[11–14] and animal contacts.[9]

The Health Population Africa (HPAfrica) study will be conducted in Burkina Faso, the Democratic Republic of the Congo, Ethiopia, Ghana, Madagascar and Nigeria. Experiences from the Typhoid Fever Surveillance in Africa Programme (TSAP) will be transferred to the Severe Typhoid in Africa (SETA) programme.[15 16] In this programme, standardised, healthcare facility-based fever surveillance is performed at selected study sites in participating countries over a 2-year period.[17] SETA primarily aims to systematically collect information on the burden of severe *Salmonella* infections. Subjects living in defined catchment areas with either an acute fever episode, a fever history of ≥3 consecutive days, clinically suspected typhoid fever or clinically diagnosed gastrointestinal perforations due to typhoid fever will be eligible. In addition, the severity of *Salmonella*-caused illnesses and the natural history of *Salmonella* infection will be recorded with further analyses on host immune responses and chronic carriage.

### Objectives and outcomes

The goal of the HPAfrica study is to assess the *generic* and *actual* age-stratified and sex-stratified health behaviour. *Generic* health behaviour pertains to behaviour in conditions associated with fever and other signs and symptoms unrelated to onsets, whereas *actual* health behaviour is related to onsets (see online supplementary appendix 1; online supplementary appendix 2: Form 4 Part A, Form 5; online supplementary appendix 3). These data will be used to adjust age-stratified and sex-stratified measures of disease occurrence like incidences of infectious pathogens in the proportion of a study population not captured by the SETA surveillance programme. Data documented for various signs and symptoms or a combination of these will be used for a gradual classification of healthcare utilisation based on disease severity.

The HPAfrica study will, moreover, collect data on healthcare-associated factors such as travel modalities, possession of a health insurance, cultural/ethnographic factors, immunisation status among children[18 19] and the frequency of occurrence and individual perception of selected diseases (see online supplementary appendix 2/online supplementary appendix 3: Form 4 Part B; online supplementary appendix 4: Form 4 Part B). Demographic data (online supplementary appendix 2: Form 3 Part A; online supplementary appendix 3: Form 3 Part A) will be used for age-stratification and sex-stratification of the population surveyed, which is required to compute adjustment factors for measures of disease occurrence. Socioeconomic data (online supplementary appendix 2: Form 3 Part B; online supplementary appendix 3: Form 3 Part B) will be used to calculate wealth indices for descriptive and analytical approaches. Information on hygiene and sanitation facilities (see online supplementary appendix 2: Form 3 Part C; online supplementary appendix 3: Form 3 Part C) may provide a better understanding of the occurrence and frequency of pathogens identified among study populations.

## METHODS

### Study sites

The boundaries of each site will be defined using pre-existing information set by statistical authorities or ministries of health (table 1) coupled with open-source high-resolution geospatial data. A retrospective review of records of SETA recruitment healthcare facilities from the past 2 years will support the redefining of site limits. It is expected that this review will reveal the residences or at least the broader administrative residential areas such as communities, districts or villages of patients who sought healthcare for any reason. Maps and satellite imagery may be used to better visualise patients' residences and the overall boundaries, including the geographically or administratively defined subareas, census enumeration areas or strata, of each site. Where boundaries cannot be clearly defined, additional factors such as the distance to a recruitment healthcare facility may be considered.

### Sample size

All available sources will be used to gather most up-to-date age-stratified and sex-stratified population data by study site and its administratively or geographically defined subareas or strata. Strata will be defined as the smallest administrative unit as published by the census of a participating country. Households within strata are expected to be approximately homogeneously distributed. Population data sources for counts per stratum may include latest demographic information from census summary data or from a Demographic Surveillance System (DSS)/Health and Demographic Surveillance System (HDSS). Population summary figures and population growth factors, if available, may be an additional data source. For sites with outdated or unavailable population counts, open-access sampling tools coupled with population data sources like

**Table 1**  Background data of participating study sites of target countries

| Country | Study site | Setting* | Approximate total site population (year) | Approximate age-stratified site population | | | | Approximate sex-stratified site population | |
|---|---|---|---|---|---|---|---|---|---|
| | | | | <2 y | ≥2 y to <5 y | ≥5 y to <15 y | ≥15 y | Male | Female |
| Burkina Faso | Nioko II† | Urban | 19251 (2017) | 2394 | 1977 | 4657 | 10223 | 9321 | 9930 |
| | Polesgo† | Rural | 7897 (2017) | 934 | 893 | 1808 | 4262 | 3856 | 4041 |
| | Ouagadougou† | Urban | 2 532 311 (2015) | 421429 | | 693723 | 1 417 159 | 1 271 302 | 1 261 009 |
| The Democratic Republic of the Congo | Kisantu‡ | Urban | 291 252 (2017) | 48043 | | 243209 | | 91 598*** | 99 231*** |
| Ethiopia | Wolayita Zone§ | Urban/Rural | 1 968 735 (2017) | 100800 | 206520 | 635015 | 1 026 300 | 964577 | 1 004 058 |
| | Wolayita Sodo§ | Urban/Rural | 120288 (2017) | 6161 | 12617 | 38807 | 62703 | 59898 | 60390 |
| | East Shewa and Arsi Zone§ | Urban/Rural | 3 249 722 (2017) | 104408 | 422939 | 1 016 917 | 1 705 458 | 1 671 699 | 1 578 023 |
| | Adama Wenji§ | Urban/Rural | 53540 (2017) | 3330 | 7204 | 17028 | 25978 | 27199 | 26341 |
| Ghana | Asante Akim North and Central¶ | Urban/Rural | 140694 (2010) | 11606 | 8363 | 35618 | 85107 | 67673 | 73021 |
| | Kumasi (Metropolis)¶** | Urban | 1 730 249 (2010) | 52516 | 178575 | 421834 | 1 077 324 | 826479 | 903770 |
| Madagascar | Antananarivo (Renivohitra)†† | Urban | 1 247 025 (2009) | NA¶¶ | NA¶¶ | NA¶¶ | NA¶¶ | NA¶¶ | NA¶¶ |
| | Imerintsiatosika‡‡ | Rural | 44669 (2016) | 3582 | 4449 | 7610 | 29028 | NA§ | NA§ |
| Nigeria | Ibadan§§ | Urban | 1 343 147 (2006) | 176110 | | 305656 | 861381 | 661818 | 681329 |

Table 1 shows population data that were available at the time of the HPAfrica protocol writing; population data and boundaries of geographically and/or administratively defined study sites may be subject to changes during the course of the study.

*The classification of sites by country is based on best local knowledge.

†http://www.insd.bf/n/; http://www.indepth-network.org/member-centres/ouagadougou-hdss.

‡The Democratic Republic of the Congo (Kisantu Central Health Zone Office report, 2016).

§Ethiopia (Health Management Information System of the Ethiopian Ministry of Health) (zonal and district health offices).

¶Ghana (Ghana Statistical Service, 2010 Population and Housing Census, Asante Akim Central Municipality).

**Ghana Statistical Service, 2010 Population and Housing Census, Summary report of final results.

††Madagascar (Population par Fokotany selon la declaration des Chefs Fokotany. Source: Donnee de la cartographie censitaire mises-a-jour en juillet 2009—INSTAT/DDSS; University of Antananarivo).

§§Nigeria (Federal Republic of Nigeria 2006 Population and Housing Census (Table DS5), National Population Commission, Abuja, Nigeria).

¶¶Population data not available at the time of protocol writing.

*** The sex-stratification is based on a total population of 190 829 applies to both, the male and female approximate sex-stratified site population for Kisantu, DRC.

NA, not available; y, years.

density-based gridded population data estimates from WorldPop may also be used.[20–25]

All strata per study site will be included. A household as defined for the HPAfrica study constitutes the primary sampling unit (PSU). The minimum number of randomly selected households to be interviewed will be calculated using the precision-based equation (Equation-I) assuming an infinite population and considering a design effect (DEFF),[26–28] including its accompanying assumptions.

### Equation-I: precision-based sample size calculation by study site for an infinite population[26–32]

$$n_0 = DEFF \cdot \left[ z_{1-\frac{\alpha}{2}}^2 \cdot \frac{4 \cdot p \cdot (1-p)}{d^2} \right]$$

#### Assumptions Equation-I

$n_0$: minimum total number of households to be interviewed in a study area assuming an infinite population; $DEFF$: design effect (set at 1.5), ≤1.0=negative correlation of the outcome(s) of interest between household members, 1.0=no correlation of the outcome(s) of interest between household members, ≥1.0=positive correlation of the outcome(s) of interest between household members; $z_{1-\frac{\alpha}{2}}$: normal deviation corresponding to a 95% CI (1.96 for α of 0.05); $d$: precision (acceptable error), point estimation, set at 0.2; $p$: proportion of the study population expected to visit a recruitment healthcare facility for conditions associated with fever and other signs and symptoms (proportion captured), set at 0.2; $1 − p$: proportion of the study population expected to not visit a recruitment healthcare facility for conditions associated with fever and other signs and symptoms (proportion not captured).

The design effect is defined as an adjustment factor for the natural clustering of health behaviour as the main outcome of interest among household members. It accounts for greater statistical variance and, therefore, lower precision compared with simple random sampling.[15 30 33 34] For HPAfrica, the DEFF may be set conservatively at 1.5 based on an estimate of 1.42 resulting from an average household size of seven and an intracluster correlation coefficient (ICC) of 0.07 from the TSAP study.[15 31 32 35] A proportion $p$ of 0.2 may be assumed if no other estimates are available, or $p$ may be based on more precise information available by site or on experiences taken from the TSAP study (table 2): Madagascar, Isotry: p=0.01, Burkina Faso, Polesgo: p=0.9.[36] Combining all assumptions into Equation-I will result in a total minimum number ($n_0$) of 92 household interviews per site.

In addition, $n_0$ will be accounted for the age distribution and sex distribution of a study population reflected by $q$, the assumed minimum proportion of stratification per age group (<5 years, ≥5 to <15 years, ≥15 years). Factor $q$ may be arbitrarily set at 0.2 or may be based on previous experiences: Ethiopia, Butajira: $q$=0.09, Burkina Faso, Polesgo: $q$=0.23.[15] Applying $q$ to Equation-II for a binomial distribution, including its accompanying assumptions, will result in the age-stratified minimum number of household interviews ($\bar{n}_0$) of 461 (Equation-II).

### Equation-II: precision-based sample size calculation by study site for an infinite population considering age-stratification and population-weight by subarea or stratum

$$\bar{n}_0 = \left( DEFF \cdot \left[ z_{1-\frac{\alpha}{2}}^2 \cdot \frac{4 \cdot p \cdot (1-p)}{d^2} \right] \right) / q \qquad \bar{n}_{0w} = \bar{n}_0 \cdot (n/N)$$

**Table 2** Sample sizes considering infinite population, differing estimates for $p$, age-stratification and loss to follow-up by study site applying Equation-II

| Proportion ($p$) | Total minimum number of households without DEFF | Total minimum number of households ($n_0$) with DEFF=1.5 | Total minimum number of households ($\bar{n}_0$) with DEFF=1.5 and $q$=0.2 | $\bar{n}_0$ adjusted for loss to follow-up |
|---|---|---|---|---|
| 0.1 | 35 | 52 | 259 | 311 |
| 0.2 | 62 | 92 | 461 | 553 |
| 0.3 | 81 | 121 | 605 | 726 |
| 0.4 | 92 | 138 | 691 | 830 |
| 0.5 | 96 | 144 | 720 | 864 |
| 0.6 | 92 | 138 | 691 | 830 |
| 0.7 | 81 | 121 | 605 | 726 |
| 0.8 | 62 | 92 | 461 | 553 |
| 0.9 | 35 | 52 | 259 | 311 |

$p$, proportion of the study population expected to visit a recruitment healthcare facility for conditions associated with fever and other signs and symptoms (proportion captured), set at 0.2; $n_0$, total minimum number of households to be interviewed in a study area assuming an infinite population;

$\bar{n}_0$, total minimum number of households to be interviewed in the study area assuming an infinite population and age-stratification;

$q$, minimum proportion of stratification per age group, set at 0.2; DEFF, design effect, set at 1.5.

## Assumptions Equation-II

$\bar{n_0}$: minimum total number of households to be interviewed in the study area assuming an infinite population and age-stratification; $\bar{n_{0w}}$: minimum number of households to be interviewed by subarea or stratum; $n/N$: population size of subarea or stratum divided by the population size of the total study area; *DEFF*: design effect (set at 1.5); $z_{1-\frac{\alpha}{2}}$: normal deviation corresponding to a 95% CI (1.96 for $\alpha$ of 0.05); *d*: precision (acceptable error), point estimation, set at 0.2; *p*: proportion of the study population expected to visit a recruitment healthcare facility for conditions associated with fever and other signs and symptoms (proportion captured), set at 0.2; $1-p$: proportion of the study population expected to not visit a recruitment healthcare facility for conditions associated with fever and other signs and symptoms (proportion not captured); *q*: the minimum proportion of stratification per age group, set at 0.2.

Health behaviour will be assessed twice at the same households if possible or at alternative households in case the study population is unstable due to migration and/or high birth/death rates. Additional households, assumed to be 20% on average (table 2) may be added to account for possible loss to follow-up between the two assessments. Eventually, population-weighted stratified sampling by the strata population proportion ($n/N$) to account for unequal selection probability due to heterogenous population distributions between strata (population-weight ($n/N$) by stratum) will be applied to $\bar{n_0}$ of 553.[37 38]

### Sampling with and without population sampling frame

If a comprehensive up-to-date population sampling frame or household list exists through DSS/HDSS or census,[39 40] computerised selection of households as PSUs will be performed using SAS (V.9.4, SAS Institute, Cary, North Carolina, USA) applying serial simple random sampling without replacement weighted by strata population proportion ($n/N$). Selected households labelled with identifiers are visualised using DSS/HDSS or census tools. In case of refusal, absence of a respondent or locating of an abandoned household, interviewers are trained to visit an alternative household closest to the right or left side of an original household. This follows the principle of closest proximity as seen in previous research.[41–44] To limit, furthermore, chances of unequal selection probability, thus chances of selection bias by interviewers, we prechoose an arbitrary number of five alternative households from the population sampling frame that require to be in closest proximity to the original household rather than applying for instance a lottery-based or clockwise spiral outward selection.[37 41 44] In addition, scoring from one (=closest) to five (=furthest) is applied to alternative households and interviewers are instructed to strictly follow the scoring sequence during the household selection procedure. Compliance of interviewers with the rules given for selecting original and alternative households will be verified on a daily basis. Deviations will be

investigated and retraining provided to interviewers as required.

If no population sampling frame exists, we will apply a stratified spatial sampling technique weighted by the strata population proportion ($n/N$).[37] Guided by previous research conducted, random spatial points are generated using the Aeronautical Reconnaissance Coverage Geographic Information System (ArcGIS, V.10.2; Redlands, California, USA) random point generation tool.[21 23 37 40 43 45–51] The tool randomly places a number of points inside the feature of a polygon that corresponds to an administrative subarea or stratum. X and Y coordinates define one point until the calculated sample size by stratum is reached. ArcGIS-generated points are converted into an image file of the Keyhole Markup Language or Keyhole Markup language Zipped format and imported into Google Earth Pro. A unique numerical identifier and geographic coordinates are assigned to each point. Online supplementary appendix 4 illustrates our approach using examples from Ghana and Madagascar. Spatial points labelled with identifiers are pictured on poster-sized printouts of (60x60 to 60x90 cm) of Google Earth Pro satellite imagery with high resolution or alternatively using an open-source application for offline/online automated navigation and mapping to locate points on the ground. Global Positioning System receivers (Garmin-eTrex; Garmin, Lenexa, Kansas, USA) allow verifying the locations. The receivers will be positioned closely to a located point, in a static position and an open area to assure barrier-free reading. Interviewers assess spatial points for the presence of a household as the PSUs.[24] In case of refusal, absence of a respondent or locating of a non-residential point, interviewers are trained to visit a preselected alternative spatial point closest to the right or left side of an original point and assess it for the presence of a household.[41–43 52] To limit chances of unequal selection probability, we preselect at least five alternative spatial points in closest proximity to the original point that represent a structure of appropriate size and rectangular or square shape from the Google Earth Pro satellite imagery using the in-built distance measurement tool. Same as for sites with a population sampling frame, scoring from one to five is applied to alternative spatial points and interviewers are instructed to follow the scoring sequence. Compliance of interviewers with the rules given for selecting original and alternative households will be verified on a daily basis. Deviations will be investigated and retraining provided to interviewers as required. In the event that a spatial point is placed in equidistance to two (or more) structures, the interviewer skips the original spatial point and chooses a preselected alternative spatial point following the scoring sequence. In case two (or more) original spatial points are placed on the same structure, the interviewer assesses the structure for the presence of a household and chooses a second (or more) preselected alternative spatial point following the scoring sequence.

Irrespective of the presence or absence of a population sampling frame, a single-family, single-household will be

 

approached directly for study participation. However, one household only is enrolled at a multifamily, single-story structure based on following procedures: the interviewer enters the structure and assesses the total number of households; the first household on the right/left side (depending on the inner construction) of the structure is approached; if the first household does not participate, the second household on the right/left side of the same structure is approached. The interviewer continues until one household by structure is enrolled. Similarly, one household only is enrolled at a multifamily, multistory structure. After assessing the total number of households present, the interviewer approaches the first household nearest to the entrance; if the first household does not participate, the second household closest to the entrance on the same/subsequent floor (depending on the inner construction) is approached; the interviewer continues until one household by structure is enrolled. The interviewer chooses a preselected alternative spatial point following the scoring sequence in case no household can be enrolled.

### Frequency of data collection

Residential points will be visited twice for household interviews during the SETA programme to assess seasonal influences on health behaviour.[53–57] Interviews will take place in differing seasons—one at the end of the dry season or at least 1 month after its beginning and one towards the end of the wet season or at least 1 month after its beginning (table 3). Currently, the HPAfrica study is ongoing in Burkina Faso, Ghana and Madagascar and is anticipated to be continued in the remaining countries during the upcoming months.

### Inclusion and exclusion criteria

Household members of all ages and both sexes living in a study area at the day of the interview will be eligible for inclusion. A household will be excluded if the designated respondent declines participation or is unavailable after three consecutive visits. All visitors and individuals with unknown residence or residence outside the study area will be excluded.

**Table 3** Seasonality in participating countries

| Country | Period of wet season | Period of dry season |
|---|---|---|
| Burkina Faso | May/June–September/October | October/November–April/May |
| The Democratic Republic of the Congo | November–March | April–October |
| Ethiopia | June–August | October–May |
| Ghana | April–July, September–November | December–March |
| Madagascar | November–March/April | April/May–October |
| Nigeria | May–August | October–April |

The seasonality by country is based on best local knowledge.

For the purpose of this investigation, a household is defined as a person or a group of related or unrelated persons living in the same dwelling unit, acknowledging one adult individual as the household head, sharing the same housekeeping arrangements and independently procuring food and other essential for living.[36]

The interview will exclusively be held with the respondent, who is an adult household member at the country-specific legal age of majority. This person may be identified as decision-maker by members of the same household and serves as a proxy for an entire household.[15] Further relevant definitions are explained in online supplementary appendix 1 and online supplementary appendix 3.

## DATA COLLECTION

On-site interviewers will be trained on locating the geographic points, identification of respondents, informed consent procedures with emphasis on voluntary participation and the deployment of standardised, pretested study forms (see online supplementary appendix 2 and online supplementary appendix 3) prior to the initiation of HPAfrica.[58] Informed consent and study forms will be forward-translated into the country-specific official language by two independent bilingual translators fluent in English and native speakers of the target language who are familiar with the concept and terminology of the forms. Both translations will be compared for discrepancies together with the translators and a coordinator, and consensus will be sought. The synthesised forward translation will be back-translated from the target language into English by two independent translators blinded to the original forms. Consensus on discrepancies of translations and the original forms will be sought together with the translators and a coordinator. Translated forms will be pilot-tested among a convenience sample of households of the target population prior to their finalisation to assure cross-cultural comprehensibility and semantic, idiomatic, experiential and conceptual equivalence. A translation report will be prepared.[58–60]

## DATA MANAGEMENT

Management of data will depend on the mode of collection, which will be primarily electronic rather than paper-based. Past experiences showed following disadvantages of paper-based data collection: a high probability of errors while filling study forms, and the need of subsequent data digitalisation, which is an extra, labour-intensive and time-intensive and error-prone step. Electronic data capturing using an application for a mobile device like smartphone or tablet minimises these limiting factors to the data quality.

There may be a pilot period during which both paper-based and paperless data will be collected prior to full implementation of electronic data collection. The latter

uses the Android application 'HPA Collect' (Google Android 5.0.1 9API 23; available at https://play.google.com/store/apps/details?id=anint.ivi.hpa) and the platform 'HPA Web' (CSS, JavaScript and JSP; available at http://hpa.ivi.int/), which are accompanied by a variable dictionary containing codes, meaning and properties of variables in at least English and French. Both, 'HPA Collect' and 'HPA Web', will be designed specifically for the purpose of this study. IVI's expertise in computer engineering allows the fundamental construction, configuration and development of components required. Different from many open-source programs, a server installation feature enables the storage of all original data collected on the institutional server besides extra features for deep-error checking, logical relations between variables and forms, search functions and analysing and displaying of data collected. Stable wireless internet connectivity using default browsers is required to assure a contemporaneous, attributable, original and accurate synchronisation of data between 'HPA Collect' and 'HPA Web'. Access to both 'HPA Collect' and 'HPA Web' is password-protected. Online supplementary appendix 5 displays the system diagram, the features of which are discussd below.

### Structure or features of 'HPA Collect'

► *User management:* a simple user interface provides different privileges for users to enter, edit (except for study labels) and export or transfer data. It establishes an audit trail that records when users enter data into the server.
► *Data input:* data are entered directly into the smartphone or tablet device. Entry fields limit possible answers to plausible ones only by offering different input methods:
  – Edit text: free text;
  – Radio group/multiclick button: single choice;
  – Check box: multiple choice;
  – Date and time picker: entering date and time;
  – Spinner: drop-down button.
► Additional features allow the surveyor to save time by transferring data collected at an earlier stage to data fields required at a later stage of the interview by performing plausibility checks for diverse data input fields, and by easily recognising skip patterns. Error checks, missing data notifications, code errors and logical errors are also included.
► *Data view:* raw, originally entered data can be seen in the same, human-readable and machine-readable format, namely the JSON format, as they will be transferred to a database server (Windows, MySQL). Additional functions like 'clear' to clear all records, 'reset' to return back to the main data view page and 'search' to search for a study label are available.
► *Export to local:* all data entered will be backed-up three-fold on a public storage folder of the mobile device as JSON and txt formats. Txt files contain data of each individual form. One version of JSON files contain data by each individual form and a second version of JSON files contain all data of a household enrolled. Data in the JSON format can be extracted by USB or wireless internet connection.
► *Report:* this feature allows a user to report any query to the leading 'HPA Collect' administrator at IVI by email or a third-party application.
► *Settings:* the ability to switch the system language between English and French is provided.
► *About:* the currently installed version of 'HPA Collect' is displayed.
► *Export:* all entered data will be sent to the IVI server through wireless internet connection. Each form record having a unique study label of the household enrolled is transformed into a SQL insertion query and inserted onto the server only if there is no duplicated study label found. Insertion of data will be ignored in case of duplicated study labels

### Structure or features of 'HPA Web'

► *Home:* n introduction to the platform is given and a tutorial in French and English will be accessible.
► *Form view:* the study forms are displayed separately showing every question and the corresponding input data as defined variables.
► *Form edit:* this feature allows the correction of the uploaded data. It is only accessible to data managers in each participating country and at IVI.
► *Search:* specific study labels given to each study form can be filtered.
► *Variable dictionary:* the variable dictionary can be downloaded.
► *Export:* data can be converted from the server into a transferrable excel file.
► *Contact us*: Quick and easy correspondence between users and the data manager at IVI is given.

Data will be reviewed by key study personnel on a day-to-day basis and checked for consistency and accuracy prior to data analysis. The quality of data may be checked by selecting a subset of 5% for validation against the original, paper-based data if possible.[58] Names of household members will not be linked to study forms of HPAfrica and will not be recorded in the database in order to ensure confidentiality. Access to the database will be restricted to authorised study personnel only and data will be kept in a locked, protected location. Periodic interim backup files and at least three final backup files of the database stored in a secure, locked place will avoid loss of data and ensure data safety. All data will be stored for a minimum of 5 years.

### Data analysis
Combined sampling weights will be computed to adjust for possible differences in selection probabilities and to increase the accuracy of study estimates.[22 61] Weights are generated in a multilevel approach to consider selection probabilities of households, inverse weights within households (ie, multifamily single-story structure, multifamily multistory structure) and inverse weights within

household members participating in the HPAfrica study.[21 43] Open-source tools, census data and HPAfrica-generated data such as the population size, the number of households or structures, the number of households within multifamily/multistory structures and the household size by stratum will be used to compute sampling weights.[62 63]

Descriptive analyses such as absolute and relative frequencies, medians or arithmetic means will be conducted to construct a contemporary baseline population description. Basic analyses will include the calculation of age-stratified and sex-stratified proportions of *generic* and *actual* health behaviour, including healthcare utilisation, for various signs and symptoms or combinations of signs and symptoms. These analyses will allow for a gradual classification based on disease severity, including 95% CI. Health behaviour will be adjusted for actual clustering among household members. Most up-to-date stratified population data by site and demographic data derived from HPAfrica will be used to generate the proportions for the adjustment of incidence calculations.

Socioeconomic information collected will be used to construct wealth indices of studied populations by applying factor analyses based on the principal component method. In addition, bivariate and multivariate regression analyses will be performed to identify potential confounding factors or effect modification using factor scores.

Bivariate and multivariate regression analyses will be carried out using factor scores for the assessment of the following variables:

► Socioeconomic data such as education, occupation,[64] health insurance, housing and household assets.
► Hygiene and sanitation such as toilet access, water sources, food handling, waste disposal and animal contacts.
► Health behaviour: *actual* (age-stratified and sex-stratified, severity, frequency), *generic* (age-stratified and sex-stratified, severity, rating by choice).
► Travel modalities to health facilities, age-stratified.
► Seasonality related to hygiene and sanitation, health behaviour, travel modalities.
► Cultural/religious/ethnographic factors, age-stratified.
► Vaccination status among children.[18 19]
► Knowledge and perception of common diseases.

## Ethical considerations
All study participants are required to provide written informed consent. There are no direct benefits for enrolled households and their members. Indirect benefits for the general population at sites will be the updated information on the burden of communicable diseases, including factors influencing disease transmission and infection. The HPAfrica study generates evidence to support decision-makers on the introduction of appropriate measures for disease prevention and control.

This protocol undergoes annual review and renewal by ethics committees of collaborating institutions (Ethics References are listed in online supplementary appendix 1).

During and after the HPAfrica study, all data of enrolled subjects will be kept in strict confidence and will not be disclosed to a third party by any member of the research team. Password protection of devices and the database is used for strict confidence. All paper-based data (consent forms) will be kept in a secure, locked location. Confidential information stored on computers and paper-based data sources will only be made available to co-investigators and IVI staff directly involved in study activities of HPAfrica.

## Patient and public involvement
No patients but household members in selected study sites will be involved. They did not participate in the development of the research question, outcome measures and the study design. However, we attempt to balance well between answering our study objectives and benefiting the general population at the sites by disseminating information on the disease burden and factors influencing disease transmission and infection derived from our study. Household members will contribute to the study conduct by direct participating in the gathering of data. They will not be involved in recruitment procedures. The households selected for participation will be identified purely by chance due to the random sampling applied. Study findings will be disseminated to study populations directly through collaborating institutions and indirectly through publications in peer-reviewed journals.

## Dissemination
We plan to disseminate the study findings by reporting descriptive as well as analytic and stratified data through publications in peer-reviewed journals and collaborating institutions.

**Author affiliations**
[1]International Vaccine Institute, Gwanak-gu, Seoul, Republic of Korea
[2]Kumasi Centre for Collaborative Research in Tropical Medicine, Kwame Nkrumah University of Science and Technology (KNUST), Ghana, Africa
[3]School of Public Health, Kwame Nkrumah University of Science and Technology (KNUST), Kumasi, Ghana
[4]Global Health Institute, Emory University, Atlanta, Georgia, USA
[5]Infectious Diseases and Geographic Medicine, Stanford University, Stanford, California, USA
[6]Service de Microbiologie, Cliniques Universitaires de Kinshasa, Kinshasa, Democratic Republic of the Congo
[7]Institut National de Recherche Biomédicales, Kinshasa, Democratic Republic of the Congo
[8]Department of Pharmaceutical Microbiology, Faculty of Pharmacy, University of Ibadan, Ibadan, Nigeria
[9]Institut Supérieur des Sciences de la Population, University of Ouagadougou, Ouagadougou, Burkina Faso
[10]Armauer Hansen Research Institute, Addis Ababa, Ethiopia
[11]University of Antananarivo, Antananarivo, Madagascar
[12]Faculty of Medicine, Duy Tan University, Da Nang, Vietnam
[13]Institute of Tropical Medicine, Eberhard Karls University, Tübingen, Germany
[14]The Department of Medicine, The University of Cambridge, Cambridge, UK
[15]Swiss Tropical and Public HealthInstitute (Swiss TPH), Basel, Switzerland

[16]University of Basel, Basel, Switzerland

**Acknowledgements** The authors would like to thank David Kauderer for the support and advice in computer engineering, Dr Paul M Tshiminyi for the support in proofreading French study materials and Soo Young Kwon and Ji Hyun Han for administrative support. The authors would also like to thank community leaders and site representatives for their assistance and facilitation in properly approaching household members for our study.

**Contributors** GDP, JKP and UP conceptualised the initial study concept of the HPAfrica study. HWS and GHJ developed the tools 'HPA Collect' and 'HPA Web' for electronic data collection with primary support of UP and AH. IO and JA supported the development and finalising of the data collection tools 'HPA Collect' and 'HPA Web'. MR and RR supported the development and finalisation of the HPAfrica study forms. The appendices were conceptualised by GDP, JKP, YC, JI, TT, AHH and UP. IO, JA, RB, LMCE, JI, HJJ, SPL, OL-M, WMW, ODM, INO, EO-D, SEP, OP, HJS, ABS, MT, TT, MR and RR supported the further development of the initial study protocol and appendices. FM, YC and CGM participated in the finalisation of the study protocol, including appendices. AH wrote the first draft of the manuscript. GDP, HWS, IO, JA, RB, LMCE, MH, JI, GHJ, HJJ, SPL, OL-M, WMW, ODM, INO, EO-D, JKP, SEP, OP, HJS, ABS, MT, TT, YC, MR, RR, CGM and FM critically reviewed and revised the manuscript draft. GDP, AHH, HWS, IO, JA, RB, LMCE, MH, JI, GHJ, HJJ, SPL, OL-M, WMW, ODM, INO, EOD, JKP, SPL, OP, HJS, ABS, MT, TT, YC, MR, RR, CGM, FM and UP agreed and approved the final manuscript as submitted.

**Funding** This study was supported by the Bill & Melinda Gates Foundation (OPP1127988). The International Vaccine Institute acknowledges its donors, including the Republic of Korea and the Swedish International Development Cooperation Agency (Sida).

**Disclaimer** The funders had no role in study design, data collection, data analysis, data interpretation or writing of the report; the conclusions and findings presented are those of the authors and do not necessarily reflect positions or policies of the Bill & Melinda Gates Foundation or the International Vaccine Institute.

**Competing interests** None declared.

**Patient consent** Not required.

**Ethics approval** Institutional Review Board of the International Vaccine Institute.

**Provenance and peer review** Not commissioned; externally peer reviewed.

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
