## [Reviewer comments · BMJ Open]

ARTICLE DETAILS

TITLE (PROVISIONAL)	The HPAfrica Study Protocol: Assessment of health behavior and population-based socio-economic, hygiene behavioral factors - a standardized repeated cross-sectional study in multiple cohorts in sub-Saharan Africa
AUTHORS	Pak, GiDeok; Haselbeck, Andrea; Seo, Hyeong-Won; Osei, Isaac; Amuasi, John; Breiman, Robert; Cruz Espinosa, Ligia Maria; Holm, Marianne; Im, Justin; Jang, Geun-Hyeog; Jeon, Hyonjin; Luby, Stephen; Lunguya-Metila, Octavie; MacWright, William; Mogeni, Ondari; Okeke, Iruka; Owusu-Dabo, Ellis; Park, Jin Kyung; Park, SeEun; Popoola, Femi; Seo, Hye-jin; Soura, Abdramane; Teferi, Mekonnen; Toy, Trevor; Chon, Yun; Rafindrakalia, Mathilde; Rakotozandrindrainy, Raphael; Meyer, Christian; Marks, Florian; Panzner, Ursula

VERSION 1 – REVIEW

REVIEWER	Sebastian Kevany University of California, USA
REVIEW RETURNED	25-Jan-2018

GENERAL COMMENTS	If I may, I am writing to say that I have now had a look at this protocol and think it looks top class in every respect. Having said that, this is a little beyond my normal field of review, so you may possibly want to consider the comments of other reviewers first which will no doubt be more detailed and comprehensive than mine! More specifically, the methods, background data, sampling, data collection and data management sections all look to be of the highest quality. One point is that, in the title, 'other population-based factors' could probably be described in a little more detail. Sorry I don't have much more to add but I hope this helps. All the best,
---

REVIEWER	Stephanie Eckman RTI International, US
REVIEW RETURNED	29-Jan-2018

GENERAL COMMENTS	I do not understand how the sampling works when there is not existing household frame. They selected random points, as shown in the appendix, but what happened next? The interviewer visited the random points (using the GPS device to get there) and then...? It's not clear how one gets from the point to the selected household. Do the authors mean that the interviewer goes to the closest household to the point? If so, that is not a simple random sample. The authors
--

	cite several papers (that I'm not familiar with) for sample selection, but they don't seem to be using any of those: "enumerate all structures (26), rastering (27), or grid cells (28, 29)." I'd like to see a cite for the technique they do use.
REVIEWER	Dana R. Thomson University of Southampton, UK
REVIEW RETURNED	17-Mar-2018
GENERAL COMMENTS	Overall, this is a well-designed and clearly explained protocol. Congratulations to this diverse, multi-national team for excellent work, especially on the tool. I have one major concern: some of the sampling techniques described are likely to result in a biased population sample. I've flagged these methods in the attached file, and suggested practical alternatives that can result in a more representative samples in these difficult-to-sample settings. The guidelines for BMJ Open require a limitations section and a STROBE checklist. Kindly add. Thank you for the opportunity to review this piece of work. I wish this team the best while implementing this important study. - The reviewer provided a marked copy with additional comments. Please contact the publisher for full details.

VERSION 1 – AUTHOR RESPONSE

Response to Reviewers (ID bmj-open-2017-021438)

Reviewer: 1

Reviewer Name: Sebastian Kevany

Institution and Country: University of California, USA

Competing Interests: None declared

If I may, I am writing to say that I have now had a look at this protocol and think it looks top class in every respect. Having said that, this is a little beyond my normal field of review, so you may possibly want to consider the comments of other reviewers first which will no doubt be more detailed and comprehensive than mine!

More specifically, the methods, background data, sampling, data collection and data management sections all look to be of the highest quality. One point is that, in the title, 'other population-based

factors' could probably be described in a little more detail. Sorry I don't have much more to add but I hope this helps. All the best,

Sebastian

Response to Reviewer 1

Dear Sebastian Kevany,

We appreciate your feedback. Following your recommendation, we revised the manuscript title to: "The

HPAfrica Study Protocol: Assessment of health behavior and population-based socio-economic, hygiene behavioral factors - a standardized repeated cross-sectional study in multiple cohorts in sub-Saharan Africa.

Reviewer: 2

Reviewer Name: Stephanie Eckman

Institution and Country: RTI International, US

Competing Interests: None declared

I do not understand how the sampling works when there is not existing household frame. They selected random points, as shown in the appendix, but what happened next? The interviewer visited the random points (using the GPS device to get there) and then...? It's not clear how one gets from the point to the selected household. Do the authors mean that the interviewer goes to the closest household to the point? If so, that is not a simple random sample. The authors cite several papers (that I'm not familiar with) for sample selection, but they don't seem to be using any of those: "enumerate all structures (26), rastering (27), or grid cells (28, 29)." I'd like to see a cite for the technique they do use.

Response to Reviewer 2

Dear Stephanie Eckmann,

Thank you for your comments on our protocol manuscript. Due to the random selection of the geographic points we cannot foresee where a point may be located on the ground. A point may or may not be located on a structure/building. In case the point is located on a structure/building, interviewers approach it for conducting the interview. In case the structure is a non-residential building or people in this household refuse to participate, interviewers apply our rule of approaching the nearest alternative

structure/building on the right or left side of the original point for the interview. In case a point may not be located on a structure, interviewers are trained on applying our rule of approaching the nearest alternative structure/building on the right or left side of the original point for the interview. We believe that our sampling strategy still is simple random sampling. Unfortunately, we don't have a citation for this method. Regarding the other sampling techniques described in the references 26-29, we considered applying them. However, due to the nature of our study sites, we think they are not applicable. Thus, we decided to apply the possibly novel method of spreading geographic random points across a study area.

Reviewer: 3

Reviewer Name: Dana R. Thomson

Institution and Country: University of Southampton, UK

Competing Interests: None declared.

Overall, this is a well-designed and clearly explained protocol. Congratulations to this diverse, multi-national team for excellent work, especially on the tool.

I have one major concern: some of the sampling techniques described are likely to result in a biased population sample. I've flagged these methods in the attached file, and suggested practical alternatives that can result in a more representative samples in these difficult-to-sample settings.

The guidelines for BMJ Open require a limitations section and a STROBE checklist. Kindly add.

Thank you for the opportunity to review this piece of work. I wish this team the best while implementing this important study.

Response to Reviewer 3

Dear Dana R. Thomson,

We appreciate your valuable feedback on our manuscript. Please find our responses to your comments below:

Line(s)	Reviewer comment	Response
---------	------------------	----------

[Based on pdf document]		
131	Which data? Should this be "These data..."	We edited the sentence to "These data..".
Table 1	Deleting age distribution Ethiopia (year) Missing ")"	We modified the table according to your recommendations.
171	Is the team familiar with gridded population data estimates? Datasets such as WorldPop provide population total estimates in small grid squares (~100m X 100m) in all countries in this study. Age-sex breakdowns are available for most countries as well. Gridded population estimates are designed to be re-aggregated to any geographic area. http://www.worldpop.org.uk/data/get_data/	We have looked into this source and have discussed the possibility of utilizing those population data. However, we are using existing administrative/geographic boundaries to define limits of our study sites up to the smallest possible level. Thus, we think that using existing country-specific population data that match the existing administrative/geographic boundaries may be an adequate approach.
197	Why isn't 461 reported here?	We included the number of interviews in the manuscript.
208	Why isn't 553 reported here?	We included the number of interviews in the manuscript.
Table 2	The text seems to describe the 2nd row of this table. Is this the target sample size per site? Why is the full table of calculations presented? Will there be different sample	Thank you for pointing this out. We refer to the 2 nd row of the table (assumed proportion p of 0.2) if no more precise estimates for the proportion p are available.

	sizes per site? Please clarify the total target	At this early state of the study we cannot
--	---	--

	sample size for this study.	assure that the sample size will be the same for all study sites. However, the minimum number of household interviews for each site will be 553.
227	Does this sample need to be representative of the population (or of the landscape)? Only enumeration of structures (26) will get you close to a population-representative sample. The methods described in references 27, 28, and 29 are spatial sampling techniques which will result in representative samples of landscapes, but not of people. Spatial sampling will over-represent larger, sparsely located buildings and under-sample smaller, densely located households, resulting in a sample biased toward wealthier households. Would a gridded population sampling method be useful here? See www.gridsample.org	During our previous multi-country Typhoid Fever Surveillance in Africa Program (TSAP) we actually applied the method of individual structure enumeration in all study sites without existing sampling frame. We realized that households located in smaller, densely areas tended to be over-represented/over-enumerated. This happened due to the fact that a household in such settings was commonly composed of several smaller structures which led to over-enumeration. However, our geospatial enumeration approach did not allow us to clearly delimit a household. We have discussed the use of gridded population sampling. However, due to the diversity in structure arrangement coupled

		with green areas, agricultural fields and surface water, we have decided not to use it. The nature of our study sites does not allow the assignment of standardized grids.
230	It is not clear if this paragraph describes (a) an additional sampling methods that should be listed in the last sentence of the previous paragraph, or (b) a 2nd stage sampling method to be paired with "rastering" and "grid cells". As mentioned before, the "rastering" and "grid cells" methods appear to be spatial sampling methods that will result in biased samples of households. If this paragraph describes an additional sampling method to "enumerate all structures", then please clarify in previous paragraph. This description was a bit difficult to follow - it seem that the team plans to use random point generation with stratification, which is great. This is because random point generation (alone) is another form of spatial sampling and will result in a biased sample.	We have revised the manuscript and have tried to clarify which method our team will be using. Yes, we are going to use random point generation with stratification in our study sites. We agree that the approach of random point generation may have limitations especially in rural, less densely populated sites. However, due to standardization we have discussed and decided to apply this method in all our sites. In case the random point did not fall on a structure/building, trained interviewers apply our rule of approaching the nearest alternative structure/building on the right or left side of the original point. Printouts of Google Earth Pro satellite imagery with high resolution up to structure level are also used to help locating alternative structures/buildings on

	Also, note, that random point generation is more arduous to implement in less densely populated settings. Half of the sampling sites have some rural population, so it is worth considering and discussing this potential	the ground.
--	--	--------------------

	limitation.	
232	Please specify what you are stratifying on here. Random point generation is another form of spatial sampling and assumes (unrealistically) that households are homogeneous distributed within the survey area. If used directly, the sample will over-represent larger, sparsely located buildings (most likely belonging to wealthier households), and under-represent smaller, densely located buildings. Researchers, however, have used random point selection of households after geographically-stratifying the survey area into regions of low-to-high population density. They generally sample from these geographic strata in proportion to the total population per stratum. WorldPop population estimates for 100m grid cells might be helpful here to estimate the total population	Thank you for your comments. We edited the manuscript to better clarify that we are addressing population-weighted stratification of the study area for our random point selection.

	per geographic strata. Example of spatial sampling methods applied to population survey: http://www.jstor.org/stable/40230993 WorldPop population estimates: http://www.worldpop.org.uk/ Possible alternative tool to select a representative household survey in the absence of a census/official population sample frame: www.gridsample.org or GridSample R algorithm https://doi.org/10.1186/s12942-017-0098-4	
273	How is an "adult" defined here? Age 15+ (Demographic and Health Surveys), or age 18+, etc?	An adult is defined based on the country-specific legal age of majority which may differ by country (see ll. 270-271).
295/296	move to reference list	These are direct links to our tools, which is why we would like to keep them in the main text.
356	What is this assumption? Aren't you collecting data about everyone in the sampled households? If so, why not use the measured clustering effect?	Thank you for pointing this out. We have corrected and revised the manuscript accordingly.
368	This reference is for occupation categories. Should "income" be called "occupation" here?	We have corrected this in the manuscript.
435	Kindly include a STROBE checklist:	We included a STROBE checklist in

(App.1)	https://www.strobe-statement.org/index.php?id=strobe-home	Appendix 1.
459	Please follow the formatting guidelines for electronic references: http://authors.bmj.com/writing-and-formatting/formatting-your-paper/	Thank you. We will revise the references following the guidelines.

VERSION 2 – REVIEW

REVIEWER	Stephanie Eckman RTI International, Washington DC, USA
REVIEW RETURNED	19-Apr-2018

GENERAL COMMENTS	The authors have not sufficiently improved the papers from the first round. The paper still does not include any discussion of why they chose this sampling method or references to other studies that have used it. The method they are using is called "in-field reverse geocoding". It is not a probability of selection method, because the houses are not selected with equal probabilities, yet the method produces no probabilities or weights. It is clear that in this method, houses that are more isolated, or larger, will have a larger probability of selection. I notice the 3rd reviewer also brought up these concerns and they were not sufficiently addressed in the revised manuscript. S/he also provided a reference to Kumar 2007 which 1) I think is a very weak paper; 2) the authors chose not to cite, even though it supports their method. Back-translation is also not considered a best practice for instrument translation. Equations 1 and 2 should not use both p (or P) and \hat{p} -- there are many other letters! Can you not get any data at all to estimate DEFF? There are also a number of free, open-source survey data collection programs -- if there was a reason to develop their own, the authors should state the reason. Survey Solutions from the World Bank is a good one.
--

	The authors should dig much more deeply into the literature on survey statistics and methodology before attempting to implement a challenging study such as this one.
--	---

REVIEWER	Dana R. Thomson University of Southampton, UK
-----------------	--

REVIEW RETURNED	11-Apr-2018
-------------

GENERAL COMMENTS	The revisions are looking better, however, the authors did not respond adequately to my core concern about bias in the sample design. Line 207: Thank you for clarifying that a minimum sample size will be selected per site assuming $p=0.2$, and that this sample size may be revised later if more precise information is available by site. I recommend adding a sentence here describing this to the reader, and giving context for Table 2. Line 226: The authors still refer to "rastering" (28) or "grid cells" (29,30) as methods considered to create a POPULATION sample frame. As I wrote previously, the cited works are spatial sampling techniques which will result in representative samples of landscapes, but not of people. The authors should either remove the references, or describe why they were deemed inappropriate for generating a population sample frame. Note, the team might instead describe here their consideration of "gridded population sampling". Here are multiple examples of this method being used in similar contexts: http://gridsample.org/img/tutorial/gridsample_tutorial_website_v1-1.pdf, https://ij-healthgeographics.biomedcentral.com/articles/10.1186/1476-072X-11-12, https://ij-healthgeographics.biomedcentral.com/articles/10.1186/s12942-017-0098-4, http://winegis.com/images/census-independent-GIS-based-sampling-strategy-for-household-surveys-plan-of-action%20removed.pdf, http://journals.plos.org/plosmedicine/article?id=10.1371/journal.pmed.1001007 Lines 228-239: Here is repeat my concerns about the sampling approach, with more detail: First, generation of N points equal to the sample size and placing them randomly over a large study area is a spatial sample. This method provides a simple random sample of space, but not a simple random sample of structures, dwellings, or households. The only exception is if the structures are all approximately the same size, and are regularly-spaced within the study area. Does this condition hold within the study areas at each site? If so, say so. In your response, you describe the study area as having a diverse arrangement of structures "coupled with green areas, agricultural fields and surface water" which indicates that this condition does not hold. When randomly or regularly placed points are generated and we select the nearest structure, we favor sparsely located structures or structures
--

	near open spaces which, particularly in towns and cities, are likelier to be occupied by wealthier households. Second, the unit of the sample is not clear - is it a structure, a dwelling, or a household? Presumably this is a household survey, and thus sample weights are needed to adjust for unequal probability of selection of households, within dwellings, within structures. Sampling weights would not be needed if all household were sampled in each structure, or if 1structure=1dwelling=1household for every household. The authors raise a good point about over-sampling multi-structure households if structures are randomly sampled. This can be addressed with sampling weights (described below). I think I understand what the authors are aiming for, and I sympathize with the challenges of selecting a robust sample when census data are outdated, inaccurate, or unavailable. Here are additional citations and more detailed recommendations that are hopefully helpful for the next round of revisions: OPTION 1: Population weighted stratified sample. This is the design that the authors propose, and it can result in an approximately unbiased sample IF the strata are sensible. The authors have not clarified what are the strata. They have also not clarified where they will obtain population counts per stratum. Please clarify both of these points. Here are two examples where this approach resulted in an approximately unbiased sample. 1) Researchers at EpiCentre used regularly placed points (instead of randomly placed points) within clusters (neighborhoods) to select nearest structures during a two-stage cluster sample. https://ete-online.biomedcentral.com/articles/10.1186/1742-7622-4-8. 2) A team in Kenya stratified their study area by census enumeration area (neighborhoods) and used regularly placed points to sample nearest structures. In both of these surveys, the essential condition held: structures were of similar-size and had approximately an even dispersion within the area where points were placed, and thus the survey was approximately a simple random sample of structures. If the authors can create strata that define different neighborhood types such that structures within each strata are approximately similar in size and dispersion, then the current sampling approach will be sufficiently unbiased. This is why I previously wrote "Researchers... have used random point selection of households after geographically-stratifying the survey area into regions of low-to-high population density. They generally sample from these geographic strata in proportion to the total population per stratum. WorldPop population estimates for 100m grid cells might be helpful here to estimate the total population per geographic strata." OPTION 2: Simple random sample of structures. If the team doesn't have the ability or time to delineate strata of low-to-high population density for each study site, another option is to 1) generate thousands of random points with the site coverage area, 2) review each point on satellite imagery from the first-to-last generated random point, keeping only those points which are located on top of a structure, and 3) stop the process when you reach the desired
--	---

sample size. EpiCentre is working on an easy-to-use tool called Geo Sampler to help teams perform this process - I think they are in the process of publishing that tool now, so you might contact them directly or monitor their website <http://www.epicentre.msf.org/en>. In terms of sampling probability, this approach is equivalent to generating a point for each structure in the study area based on satellite imagery and sampling points at random (like your previous study). As you mentioned, this approach over-samples households that have multiple structures. You can easily adjust for this unequal probability of selection by collecting a few pieces of information during the interview (e.g. How many structures does your household have including latrine, kitchen, sheds, and animal shelters?), and applying sampling weights during analysis.

The sampling weight is 1 over the probability of selection $w=1/P(\text{selection})$. The probability of selection is the combined probability that one of the household's structures was selected from satellite imagery $P(\text{structure})$, probability that one dwelling was selected among all dwellings in the structure $P(\text{dwelling})$, and probability that one household was selected among all households in the dwelling $P(\text{household})$. Note that $P(\text{structure})$ is inverted in this equation because multi-structure households have higher, not lower, chance of selection.

$$w = 1 / [(1 / \# \text{structures}) + (1 / \# \text{ dwellings in structure}) + (1 / \# \text{ households in dwelling})]$$

OPTION 3: Gridded Population Sampling. I'm not sure if the team fully appreciates the power of gridded population sampling, and I recommend reviewing the articles cited above re Line 226. You will find that various versions of gridded population samples have been implemented successfully in settings much like your own.

The website www.gridssample.org is the most user-friendly tool available at the moment, though as you pointed out, it is limited to predefined administrative areas. Another option which allows the team to use custom boundaries for each study site is the GridSample R algorithm <https://cran.r-project.org/package=gridssample>, which is documented in a paper <https://ij-healthgeographics.biomedcentral.com/articles/10.1186/s12942-017-0098-4>. Full disclosure - I developed this tool.

Other groups such as RTI <https://www.rti.org/impact/geosampling-survey-tool> and the World Bank <https://blogs.worldbank.org/opendata/what-can-you-do-high-resolution-population-map> are working on similar tools, however, neither have released the tools publicly yet, though you may find you have more options in tools by the time you are implementing surveys. Use of the GridSample R algorithm can result in neighborhood-sized primary sampling units (aka clusters) inside which you can apply OPTION 1, similar to EpiCentre and the Kenya teams.

Line 236: What is a "horizontal geographic coordinate"? Should "horizontal" be removed?

	Line 248: When field enumerators are responsible for identification of nearest structure, the study is susceptible to intentional or unintentional selection bias. This is the same critique of spin-the-pen and random-walk survey methods. Selection of structures should ideally be done in the office, perhaps using satellite imagery, before fieldwork. It is straight forward to select nearest structure from the office before fieldwork, and thus recommended to strengthen the protocol.
--	---

VERSION 2 – AUTHOR RESPONSE

Response to Reviewers (ID bmj-open-2017-021438.R1)

Reviewer: 2

Reviewer Name: Stephanie Eckman

Institution and Country: RTI International, Washington DC, USA

Competing Interests: None declared

The authors have not sufficiently improved the papers from the first round.

The paper still does not include any discussion of why they chose this sampling method or references to other studies that have used it. The method they are using is called "in-field reverse geocoding". It is not a probability of selection method, because the houses are not selected with equal probabilities, yet the method produces no probabilities or weights. It is clear that in this method, houses that are more isolated, or larger, will have a larger probability of selection.

I notice the 3rd reviewer also brought up these concerns and they were not sufficiently addressed in the revised manuscript. S/he also provided a reference to Kumar 2007 which 1) I think is a very weak paper; 2) the authors chose not to cite, even though it supports their method.

Response:

Thank you for your comments on the sampling methodology and for noting the term "in-field reverse geocoding". The approach we are using is following:

At most of our HPAfrica sites comprehensive data on households or property size is not available. We use the information on the most up-to-date population size of the smallest administrative units or strata by site. We calculate the number of samples required to be selected within each of the smallest unit or stratum applying population-weighted probability of the population of the whole study area. Within the boundary or polygon of each smallest administrative unit, ArcGIS creates random spatial points, which are located by the interviewers in the field. The interviewers are trained to assess these located spatial

points for the presence of a household. In case no household can be identified at the spatial point, interviewers refer to pre-selected alternative spatial points to limit chances of unequal selection probability. We explained our approach in more detail in the section “Sampling with and without population sampling frame” of the manuscript.

In addition to the revision of the corresponding sections in the manuscript we added the reference Kumar 2007 “Spatial Sampling Design for a Demographic and Health Survey”.

Back-translation is also not considered a best practice for instrument translation.

Response:

Thank you for your feedback on the issue of instrument translation. We have revised the respective section of the protocol. It reads as: “Informed consent and study forms will be forward-translated into the country-specific official language by two independent bilingual translators fluent in English and native speakers of the target language who are familiar with the concept and terminology of the forms. Both translations will be compared for discrepancies together with the translators and a coordinator, and consensus will be sought. The synthesized forward translation will be back-translated from the target language into English by two independent translators blinded to the original forms. Consensus on discrepancies of translations and the original forms will be sought together with the translators and a coordinator. Translated forms will be pilot-tested among a convenience sample of households of the target population prior to their finalization to assure cross-cultural comprehensibility and semantic, idiomatic, experiential and conceptual equivalence. A translation report will be prepared.”

Equations 1 and 2 should not use both p (or P) and \hat{p} -- there are many other letters!

Response:

Following the recommendation, we have replaced “ \hat{p} ” in equation II and in the section “sample size” of the main text with the letter “ q ” and hope that this reads better.

Can you not get any data at all to estimate DEFF?

Response:

Referring to previous experiences of our Typhoid Fever Surveillance in Africa Program (TSAP) study conducted in 10 sub-Saharan African countries, we have computed a design effect (DEFF) estimate of 1.42 assuming an average household size of seven and an Intracluster Correlation Coefficient (ICC) of 0.07. For the HPAfrica study we consider similar assumptions as for TSAP and have set the DEFF conservatively at 1.5. The main text of the section “sample size” was revised accordingly.

There are also a number of free, open-source survey data collection programs -- if there was a reason to develop their own, the authors should state the reason. Survey Solutions from the World Bank is a good one.

Response:

Thank you for your comment. At an early stage of the HPAfrica study we have considered alternative open-source data collection programs. However, we have decided to develop our own data collection and data management tools, namely "HPA Collect" and "HPA Web" for the following reasons.

- Many open-source data collection programs are web-based systems which require continuous, stable internet connection. However, instable internet access and connectivity at many of the HPAfrica sites prevent us from utilizing those programs.
- Collaborating sites want to build up capacities by using web-based as well as mobile data collection and management tools. To satisfy those needs and to foster capacity building, we decided to develop our own data system that is suitable for desktop/PC and web as well as mobile use.
- Different from many open-source programs, our data system has features that allow data storage on the institutional server for higher data safety, sophisticated deep-error and edit checks, and logical relations across study forms.

Reasons for developing our own data collection and management system are also stated in the revised manuscript in the section "data management". It reads as "Both, "HPA Collect" and "HPA Web", will be designed specifically for the purpose of this study. IVI's expertise in computer engineering allows the fundamental construction, configuration and development of components required. Different from many open-source programs, a server installation feature enables the storage of all original data collected on the institutional server besides extra features for deep-error checking, logical relations between variables and forms, search functions, and analyzing and displaying of data collected."

The authors should dig much more deeply into the literature on survey statistics and methodology before attempting to implement a challenging study such as this one.

Reviewer: 3

Reviewer Name: Dana R. Thomson

Institution and Country: University of Southampton, UK

Competing Interests: None declared.

The revisions are looking better, however, the authors did not respond adequately to my core concern about bias in the sample design.

Line 207: Thank you for clarifying that a minimum sample size will be selected per site assuming $p=0.2$, and that this sample size may be revised later if more precise information is available by site. I recommend adding a sentence here describing this to the reader, and giving context for Table 2.

Response:

Thank you for pointing this out. The protocol was revised and reads like this: "A proportion p of 0.2 may be assumed if no other estimates are available, or p may be based on more precise information available by site or on experiences taken from published data like the TSAP study (Table 2): Madagascar, Isotry: $p=0.01$, Burkina Faso, Polesgo: $p=0.926$."

Line 226: The authors still refer to "rastering" (28) or "grid cells" (29,30) as methods considered to create a POPULATION sample frame. As I wrote previously, the cited works are spatial sampling techniques which will result in representative samples of landscapes, but not of people. The authors should either remove the references, or describe why they were deemed inappropriate for generating a population sample frame.

Note, the team might instead describe here their consideration of "gridded population sampling". Here are multiple examples of this method being used in similar contexts:

http://gridsample.org/img/tutorial/gridsample_tutorial_website_v1-1.pdf,

<https://ij-healthgeographics.biomedcentral.com/articles/10.1186/1476-072X-11-12>,
<https://ijhealthgeographics>.

biomedcentral.com/articles/10.1186/s12942-017-0098-4,

<http://winegis.com/images/census-independent-GIS-based-sampling-strategy-for-household-surveysplan-of-action%20removed.pdf>, [http://journals.plos.org/plosmedicine/article?](http://journals.plos.org/plosmedicine/article?id=10.1371/journal.pmed.1001007)

[id=10.1371/journal.pmed.1001007](http://journals.plos.org/plosmedicine/article?id=10.1371/journal.pmed.1001007)

Response:

Thank you for your valuable feedback. We have removed the references and hope that the protocol reads better. Thank you for pointing out the "gridded population sampling" approach. We have read the examples that you provided with high interest and we have considered including this alternative in our protocol manuscript. We will try to explore the application of gridded population sampling specifically in sites of our multi-country study that have outdated census data (e.g., Ibadan, Nigeria; Kumasi, Ghana; Antananarivo, Madagascar).

Lines 228-239: Here is repeat my concerns about the sampling approach, with more detail:

First, generation of N points equal to the sample size and placing them randomly over a large study area is a spatial sample. This method provides a simple random sample of space, but not a simple random sample of structures, dwellings, or households. The only exception is if the structures are all approximately the same size, and are regularly-spaced within the study area. Does this condition hold within the study areas at each site? If so, say so. In your response, you describe the study area as having a diverse arrangement of structures "coupled with green areas, agricultural fields and surface water" which indicates that this condition does not hold. When randomly or regularly placed points are generated and we select the nearest structure, we favor sparsely located structures or structures near open spaces which, particularly in towns and cities, are likelier to be occupied by wealthier households.

Response:

We agree with your concerns that our method generates a sample of spatial points, but not samples of structures, dwellings or households. We have revised our manuscript to clarify when we refer to spatial points or households during the sampling procedure. We provide instructions in order to reduce selection bias by interviewers, particularly favoring selection of large and/or sparsely located structures. In case no household can be identified at the spatial point, interviewers refer to pre-selected alternative spatial points. We explained our approach in more detail in the section "Sampling with and without population sampling frame". Among others we describe in detail how we sample proportional to size (weighted by population density) to ensure that more points are placed in more densely populated areas than in less densely populated ones, to counter the favoring of more sparsely located structures in sampling.

After the first survey is completed, we will verify GPS information of spatial points enrolled. We will investigate them for accuracy prior to the conduct of the second follow-up survey.

Second, the unit of the sample is not clear - is it a structure, a dwelling, or a household? Presumably this is a household survey, and thus sample weights are needed to adjust for unequal probability of selection of households, within dwellings, within structures. Sampling weights would not be needed if all household were sampled in each structure, or if $1\text{structure}=1\text{dwelling}=1\text{household}$ for every household. The authors raise a good point about over-sampling multi-structure households if structures are randomly sampled. This can be addressed with sampling weights (described below).

Response:

Our basic sampling unit is a household. We will enroll only one household per structure (single-family single story, multi-family single story, and multi-family multi-story). We appreciate your recommendation to include the calculation of sampling weights and will include this in our analyses.

I think I understand what the authors are aiming for, and I sympathize with the challenges of selecting a robust sample when census data are outdated, inaccurate, or unavailable. Here are additional citations and more detailed recommendations that are hopefully helpful for the next round of revisions:

OPTION 1: Population weighted stratified sample. This is the design that the authors propose, and it can result in an approximately unbiased sample IF the strata are sensible. The authors have not clarified what are the strata. They have also not clarified where they will obtain population counts per stratum. Please clarify both of these points.

Here are two examples where this approach resulted in an approximately unbiased sample. 1) Researchers at EpiCentre used regularly placed points (instead of randomly placed points) within clusters (neighborhoods) to select nearest structures during a two-stage cluster sample. <https://eteonline.biomedcentral.com/articles/10.1186/1742-7622-4-8>. 2) A team in Kenya stratified their study area by census enumeration area (neighborhoods) and used regularly placed points to sample nearest structures.

In both of these surveys, the essential condition held: structures were of similar-size and had approximately an even dispersion within the area where points were placed, and thus the survey was approximately a simple random sample of structures. If the authors can create strata that define different neighborhood types such that structures within each strata are approximately similar in size and dispersion, then the current sampling approach will be sufficiently unbiased. This is why I previously wrote "Researchers... have used random point selection of households after geographically-stratifying the survey area into regions of low-to-high population density. They generally sample from these geographic strata in proportion to the total population per stratum. WorldPop population estimates for 100m grid cells might be helpful here to estimate the total population per geographic strata."

OPTION 2: Simple random sample of structures. If the team doesn't have the ability or time to delineate strata of low-to-high population density for each study site, another option is to 1) generate thousands of random points with the site coverage area, 2) review each point on satellite imagery from the first-to-last generated random point, keeping only those points which are located on top of a structure, and 3) stop the process when you reach the desired sample size. EpiCentre is working on an easy-to-use tool called

Geo Sampler to help teams perform this process - I think they are in the process of publishing that tool now, so you might contact them directly or monitor their website <http://www.epicentre.msf.org/en>. In terms of sampling probability, this approach is equivalent to generating a point for each structure in the study area based on satellite imagery and sampling points at random (like your previous study). As you mentioned, this approach over-samples households that have multiple structures. You can easily adjust for this unequal probability of selection by collecting a few pieces of information during the interview (e.g. How many structures does your household have including latrine, kitchen, sheds, and animal shelters?), and applying sampling weights during analysis.

The sampling weight is 1 over the probability of selection $w=1/P(\text{selection})$. The probability of selection is the combined probability that one of the household's structures was selected from satellite imagery $P(\text{structure})$, probability that one dwelling was selected among all dwellings in the structure $P(\text{dwelling})$, and probability that one household was selected among all households in the dwelling $P(\text{household})$. Note that $P(\text{structure})$ is inverted in this equation because multi-structure households have higher, not lower, chance of selection.

$$w = 1 / [(\#structures / 1) + (1 / \# dwellings in structure) + (1 / \# households in dwelling)]$$

OPTION 3: Gridded Population Sampling. I'm not sure if the team fully appreciates the power of gridded population sampling, and I recommend reviewing the articles cited above re Line 226. You will find that various versions of gridded population samples have been implemented successfully in settings much like your own.

The website www.gridsample.org is the most user-friendly tool available at the moment, though as you pointed out, it is limited to predefined administrative areas. Another option which allows the team to use custom boundaries for each study site is the GridSample R algorithm <https://cran.r-project.org/package=gridsample>, which is documented in a paper <https://ijhealthgeographics.biomedcentral.com/articles/10.1186/s12942-017-0098-4>. Full disclosure - I developed this tool.

Other groups such as RTI <https://www.rti.org/impact/geosampling-survey-tool> and the World Bank

<https://blogs.worldbank.org/opendata/what-can-you-do-high-resolution-population-map> are working on similar tools, however, neither have released the tools publicly yet, though you may find you have more options in tools by the time you are implementing surveys. Use of the GridSample R algorithm can result in neighborhood-sized primary sampling units (aka clusters) inside which you can apply OPTION 1, similar to EpiCentre and the Kenya teams.

Response:

We have defined a stratum as smallest geographic or administrative unit. The sources for population counts per stratum include census summary data and information of Demographic Surveillance Systems (DSS), Health and Demographic Surveillance Systems (HDSS). In addition, we will explore the availability of gridded population data (e.g., GridSample.org) for our study sites.

We find the idea to create strata that define different neighborhood types considering low-to-high population density very valuable. We will try to include data collection of a set of pre-determined area criteria, including data analyses adjusting for neighborhood types, in a pilot study to estimate the feasibility within our multi-country approach.

Relating to the suggested Option 2, we are afraid, that this is too time and resource consuming in our study sites (see population sizes in Table 1). Once the Geo Sampler is published we will consider this tool for smaller study areas possibly within this and even other projects.

Line 236: What is a "horizontal geographic coordinate"? Should "horizontal" be removed?

Response:

The term "horizontal" was removed.

Line 248: When field enumerators are responsible for identification of nearest structure, the study is susceptible to intentional or unintentional selection bias. This is the same critique of spin-the-pen and random-walk survey methods. Selection of structures should ideally be done in the office, perhaps using satellite imagery, before fieldwork. It is straight forward to select nearest structure from the office before fieldwork, and thus recommended to strengthen the protocol.

Response:

We agree that there could be potential intentional or unintentional selection bias and we implemented your recommendation. We will pre-select alternative spatial points in closest proximity to the original spatial point. Scoring of the alternative spatial points will be applied and interviewers will be instructed to follow the scoring sequence.

VERSION 3 – REVIEW

REVIEWER	Stephanie Eckman RTI International USA
REVIEW RETURNED	10-Jul-2018

GENERAL COMMENTS	Authors should acknowledge that the sample is not a probability sample -- that's ok, many surveys use nonprobability samples. All random walk samples are nonprobability. The authors STILL have not read and incorporated the basic literature on survey methods and survey statistics.
--

REVIEWER	Dana R. Thomson University of Southampton, UK
REVIEW RETURNED	13-Jul-2018

GENERAL COMMENTS	Thank you for revisiting the sample methods. I only have one major piece of feedback - I think the term PPPS is used incorrectly in a few places. PPPS refers to sampling of areal units which each have different population totals. Major issues: Line 197-200: I don't think is a fair summary of the sampling methods described below. Isn't it more accurate to say "Households will be selected by simple random sampling or by stratified spatial sampling, weighted by strata population proportion n/N..."? Line 219; The inserted text "PPPS population-weighted stratified" doesn't make sense to me. If you have a database of households, it makes sense to apply "serial simple random selection without replacement" as initially described. Don't you want to delete the inserted text? Line 233: This paragraph still describes OPTION1, a population weighted stratified sample, as described in my previous review. This is fine. But the authors still have not defined strata or how they will estimate population totals per stratum.
---

	Thus I recommend changing the opening sentence, eg "If no population sampling frame exists, we will apply a population-weighted stratified spatial sampling technique.[36]" Next describe the strata, e.g "Strata will be defined as the smallest administrative unit published by the census. Households within strata are expected to be approximately homogeneously distributed." Next describe how population count per strata will be calculated, eg "The sources for population counts per stratum will be from census summary data or Health and Demographic Surveillance Systems (HDSS). For sites with outdated or unavailable population counts, we will use WorldPop gridded population data estimates (cite: www.worldpop.org.uk/data/get_data/)." Minor comments: Line 199: Replace "inhomogeneous" with "heterogenous" Citation 36: The cited paper by Kumar defines strata by geographic covariates in an innovative way, then spatially samples households within strata. It's a bit of a stretch to say that this team is following the same study design. Though I think the methods in one or both of the below studies is similar to your own - right? (Sorry - I forgot to include the link to the Siri paper in my previous response... but I described both of these studies... they are worth reading!) Siri et al 2008: https://doi.org/10.1186/1475-2875-7-39 Grais et al 2007: https://doi.org/10.1186/1742-7622-4-8
--	--

VERSION 3 – AUTHOR RESPONSE

Reviewer: 2

Reviewer Name: Stephanie Eckman

Institution and Country: RTI International USA

Competing Interests: None declared

Authors should acknowledge that the sample is not a probability sample -- that's ok, many surveys use nonprobability samples. All random walk samples are nonprobability. The authors STILL have not read and incorporated the basic literature on survey methods and survey statistics.

Response: Thank you for pointing this out. We have revised and tried to correct this important subject throughout the respective sections of the revised manuscript. Please see also the first and second responses to reviewer 3, which address the description of our sampling methods in more detail. Likewise, we have read through and incorporated more survey-related literature. We hope this makes the manuscript overall scientifically sounder.

Reviewer: 3

Reviewer Name: Dana R. Thomson

Institution and Country: University of Southampton, UK

Competing Interests: None declared

Thank you for revisiting the sample methods. I only have one major piece of feedback - I think the term PPS is used incorrectly in a few places. PPS refers to sampling of areal units which each have different population totals.

Response: Thank you for pointing this out. We have revised the manuscript sections "sample size" and "sampling with and without population sampling frame" to clarify that we aim to account for differences in population sizes in the strata assuming that sampling is performed with the same probability. We hope both sections read better.

Major issues:

Line 197-200: I don't think is a fair summary of the sampling methods described below. Isn't it more accurate to say "Households will be selected by simple random sampling or by stratified spatial sampling, weighted by strata population proportion n/N ..."?

Response: We revised the respective sections. They read as "If a comprehensive up-to-date population sampling frame or household list exists through DSS/HDSS or census, computerized selection of households as PSUs will be performed using SAS (Statistical Analysis System, version 9.4, SAS Institute, Cary, NC) applying serial simple random sampling without replacement weighted by the strata population proportion (n/N)." and as "If no population sampling frame exists, we will apply a stratified spatial sampling technique weighted by the strata population proportion (n/N) .[36]".

Line 219; The inserted text "PPPS population-weighted stratified" doesn't make sense to me. If you have a database of households, it makes sense to apply "serial simple random selection without replacement" as initially described. Don't you want to delete the inserted text?

Response: "PPPS population-weighted stratified" was deleted.

Line 233: This paragraph still describes OPTION1, a population weighted stratified sample, as described in my previous review. This is fine. But the authors still have not defined strata or how they will estimate population totals per stratum. Thus I recommend changing the opening sentence, eg "If no population sampling frame exists, we will apply a population-weighted stratified spatial sampling technique.[36]"

Response: The sentence was revised and reads as "If no population sampling frame exists, we will apply a stratified spatial sampling technique weighted by the stratum population proportion (n/N)."

Next describe the strata, e.g "Strata will be defined as the smallest administrative unit published by the census. Households within strata are expected to be approximately homogeneously distributed."

Response: A brief description about the strata was added. It reads as "Strata will be defined as the smallest administrative unit as published by the census of a participating country. Households within strata are expected to be approximately homogeneously distributed."

Next describe how population count per strata will be calculated, eg "The sources for population counts per stratum will be from census summary data or Health and Demographic Surveillance Systems (HDSS). For sites with outdated or unavailable population counts, we will use WorldPop gridded population data estimates (cite: www.worldpop.org.uk/data/get_data/)."

Response: Thank you for your valuable recommendation. It was incorporated and reads as "Population data sources for counts per stratum may include latest demographic information from census summary data or from a Demographic Surveillance System (DSS)/Health and Demographic Surveillance System (HDSS). Population summary figures and population growth factors, if available, may be and additional data source. For sites with outdated or unavailable population counts, open-access sampling tools coupled with population data sources like density-based gridded population data estimates from WorldPop may also be used." The web link www.worldpop.org.uk/data/get_data/ was added to the list of references."

Minor comments:

Line 199: Replace "inhomogeneous" with "heterogeneous"

Response: The wording was changed as recommended.

Citation 36: The cited paper by Kumar defines strata by geographic covariates in an innovative way, then spatially samples households within strata. It's a bit of a stretch to say that this team is following the same study design. Though I think the methods in one or both of the below studies is similar to your own - right? (Sorry - I forgot to include the link to the Siri paper in my previous response... but I described both of these studies... they are worth reading!)

Siri et al 2008: <https://doi.org/10.1186/1475-2875-7-39>

Grais et al 2007: <https://doi.org/10.1186/1742-7622-4-8>

Response: The study design presented in the paper of Kumar et al. is not the same as ours. It just guided us same as the research done by Siri et al. and Grais et al.; both papers are included in the list of references as reference 37 and 40, respectively, and cited throughout the manuscript.